# Crenactin forms actin-like double helical filaments regulated by arcadin-2

Thierry Izoré, Danguole Kureisaite-Ciziene, Stephen H McLaughlin, Jan Löwe*

MRC Laboratory of Molecular Biology, Francis Crick Avenue, Cambridge, United Kingdom

**Abstract** The similarity of eukaryotic actin to crenactin, a filament-forming protein from the crenarchaeon *Pyrobaculum calidifontis* supports the theory of a common origin of Crenarchaea and Eukaryotes. Monomeric structures of crenactin and actin are similar, although their filament architectures were suggested to be different. Here we report that crenactin forms *bona fide* double helical filaments that show exceptional similarity to eukaryotic F-actin. With cryo-electron microscopy and helical reconstruction we solved the structure of the crenactin filament to 3.8 Å resolution. When forming double filaments, the 'hydrophobic plug' loop in crenactin rearranges. Arcadin-2, also encoded by the arcade gene cluster, binds tightly with its C-terminus to the hydrophobic groove of crenactin. Binding is reminiscent of eukaryotic actin modulators such as cofilin and thymosin β4 and arcadin-2 is a depolymeriser of crenactin filaments. Our work further supports the theory of shared ancestry of Eukaryotes and Crenarchaea.

## Introduction

Eukaryotic actin and non-eukaryotic actin-like proteins and their filaments fulfil diverse functions, from cellular motility and plasticity, cellular junctions, cytokinesis and phagocytosis in eukaryotic cells to cell shape maintenance, intracellular organisation and plasmid segregation in bacteria and archaea (*Pilhofer and Jensen, 2013*).

At the subunit level, all actins and actin-like proteins share a well-conserved three-dimensional structure as well as the longitudinal protofilament contact architecture that facilitates linear polymerisation into strands (protofilaments) and polymerisation-induced ATPase activation (*Dominguez and Holmes, 2011*; *Ozyamak et al., 2013b*). However, largely due to different contacts between strands, actin-like proteins evolved a wide variety of filament architectures. While actin forms right-handed, parallel and staggered filaments in all Eukaryotes (F-actin) (*Holmes et al., 1990*; *von der Ecken et al., 2015*), actin-like proteins differ in their filament architectures: MamK, required for magnetosome alignment in magnetotactic bacteria, forms right-handed filaments that have juxtaposed, non-staggered subunits (*Bergeron et al., 2016*; *Löwe et al., 2016*; *Ozyamak et al., 2013a*). ParM, making mitosis-like bipolar spindles during *E. coli* R1 plasmid segregation, produces left-handed, staggered filaments (*Bharat et al., 2015*; *Gayathri et al., 2012*). MreB, essential for cell-shape maintenance in most rod-shaped bacteria, forms the most divergent filaments when compared to F-actin, as it forms apolar, non-staggered and non-helical filaments that bind directly to membranes (*Salje et al., 2011*; *van den Ent et al., 2014*). The selective advantages that led to this diversity are not clear and will require deciphering the precise molecular mechanisms of the processes these filaments engage in.

Attempts to trace back the origins of today's eukaryotic F-actin filaments led to the discovery of crenactin found in certain organisms of the order Thermoproteales within the phylum Crenarchaeota (Crenarchaea, part of the 'TACK' superphylum) (*Ettema et al., 2011*). Together with other findings that report unique similarities between organisms of the TACK superphylum and Eukaryotes in

*For correspondence: jyl@mrc-lmb.cam.ac.uk

Competing interests: The authors declare that no competing interests exist.

cytokinesis (*Lindås et al., 2008*; *Samson et al., 2008*), membrane remodelling, cell shape determination and protein recycling, this has led to a theory of a common origin with eukaryotic cells (*Guy and Ettema, 2011*). Hence, it has been proposed that crenactin filaments share a common ancestor with F-actin.

In *Pyrobaculum calidifontis*, crenactin is encoded within the arcade cluster of genes, together with four arcadins, and has been proposed to be part of a cell-shape maintenance system (*Ettema et al., 2011*). Not much is known about the arcadins, but arcadin-4 is related by sequence to SMC-like proteins, in particular Rad50 (*Figure 1F*).

In recently reported crystal structures, crenactin formed filaments that consist of a single strand (*Izoré et al., 2014*; *Lindås et al., 2014*). The similarity of the monomer to eukaryotic actin was unprecedented, with an overall RMSD of 1.6 Å, despite sequence identity of only ~20%. The structure revealed the presence of a feature that resembles the 'hydrophobic plug', which makes inter-protofilament contacts in F-actin (*Holmes et al., 1990*; *von der Ecken et al., 2015*). The hydrophobic plug is longer in crenactin but is inserted in the same part of the fold as in actin.

Given these striking similarities between the actin and crenactin monomers, it has been puzzling that crenactin filaments were reported by electron microscopy to form single, rather than double helical, F-actin-like filaments (*Braun et al., 2015*).

Here, we present the near-atomic resolution structure of double-stranded crenactin filaments at 3.8 Å by cryoEM, revealing their close relationship to F-actin. In addition, we show that crenactin interacts with two of the arcadins, arcadin-1 and arcadin-2. Arcadin-2 depolymerises crenactin filaments by binding its C-terminus into crenactin's hydrophobic groove, a mode of action related to eukaryotic actin modulators, widening the potential evolutionary links (*Dominguez, 2004*).

## Results and discussion

### Crenactin forms double-helical filaments

Previous studies on the architecture of crenactin filaments were performed under high salt concentrations (>0.5 M KCl) (*Braun et al., 2015*; *Izoré et al., 2014*) that might not be entirely justified given *Pyrobaculum calidifontis'* environmental and laboratory growth conditions (*Amo et al., 2002*), although the intracellular osmolarity is currently not known. To exclude the possibility that such high salt concentration might have altered filament architecture, we carried out experiments in low-salt buffer (50 mM ammonium carbonate, 20 mM KCl, see Materials and methods).

We imaged filaments by cryo-electron microscopy using a 300 kV FEG microscope coupled to a direct electron detector. A total of 1474 micrographs showing long and highly contrasted filaments (*Figure 1A*) were collected under low-dose conditions with dose fractionation. From these we extracted 470,396 helical segments and performed reference-free 2D class averaging in RELION 2.0 (*Scheres, 2012*). The resulting classes showed a very regular pattern that was much more similar to the calculated re-projection of double-stranded F-actin (*von der Ecken et al., 2015*) than to that of single-stranded crenactin filaments from crystallography (*Figure 1B*) (*Izoré et al., 2014*; *Lindås et al., 2014*). To eliminate any bias from enforcing double helical symmetry during reconstruction in RELION (using an implementation of iterative helical real space reconstruction, IHRSR) (*Egelman, 2007*), we reconstructed the data into 3D using two different procedures. The first reconstruction was calculated using a double-stranded filament as the initial model and with symmetry that averages the two strands together (twist: 198.1° [equivalent to −161.9°], rise: 25.6 Å), whereas the second reconstruction was performed with a single-stranded initial model and helical parameters that symmetrise along one strand only, not averaging the two strands (twist: 36.2°, rise: 51.3 Å). This way, if the imaged filaments had been single-stranded, then the two reconstructions would have produced different results, one potentially double-stranded (and poor) and one single-stranded. We found that both reconstructions generated very similar double-helical density maps, the one with lower symmetry at slightly lower resolution, as would be expected (3.8 Å vs 4.2 Å) because of the smaller number of asymmetric units averaged (*Figure 1C*), thus unequivocally demonstrating the double-stranded nature of the crenactin filaments.

We then used the 3.8 Å density map to build an atomic model of crenactin in its filament form. For this, we started by placing the previous crystal structure in the cryoEM map (*Izoré et al., 2014*), and then manually modified and computationally refined the structure, yielding a reliable atomic

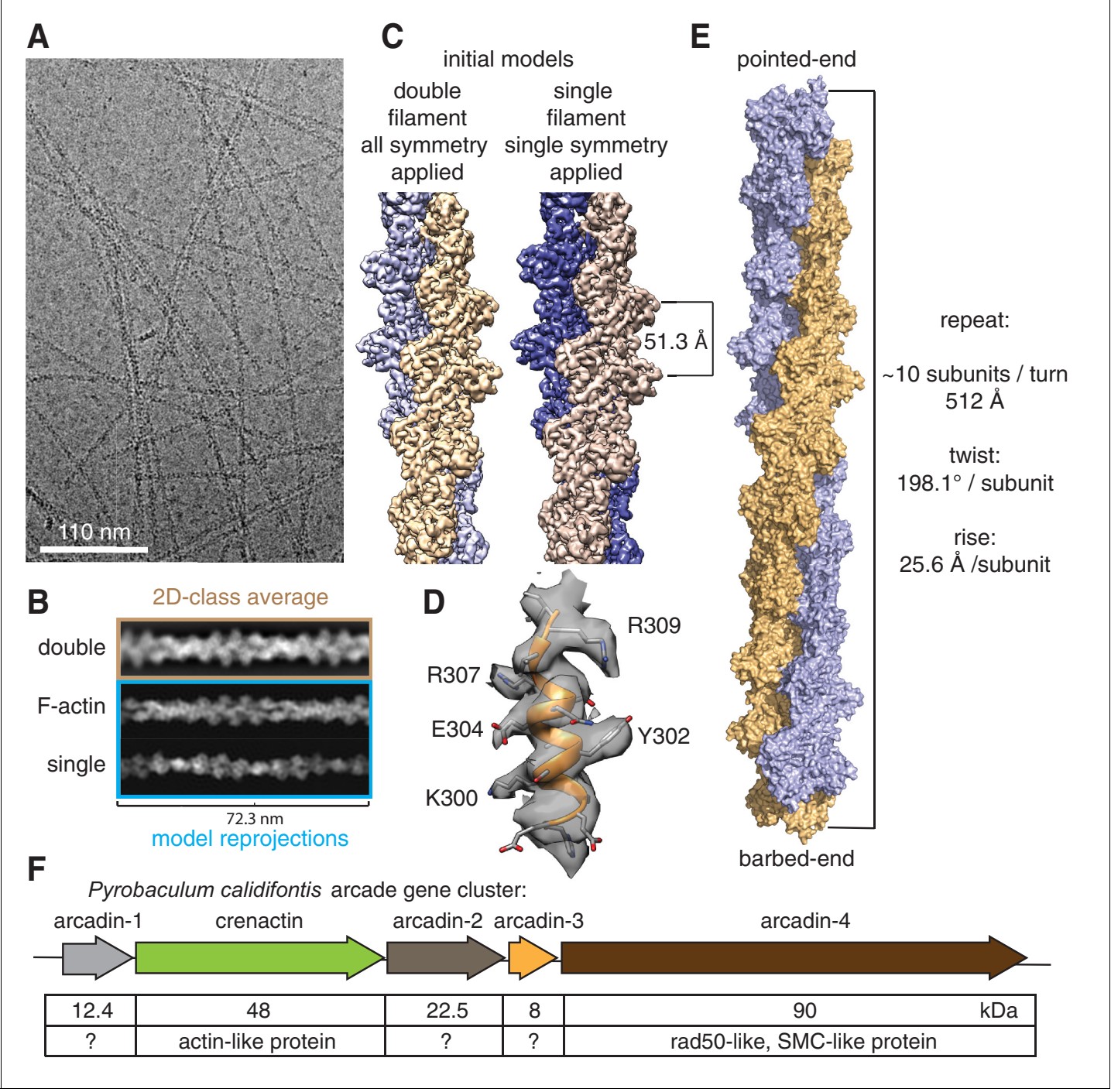

**Figure 1.** Crenactin forms double filaments. (**A**) Representative cryoEM micrograph of crenactin filaments. (**B**) Comparison between crenactin double helical 2D class average (top), re-projections of F-actin double-stranded filament structure (middle) (*von der Ecken et al., 2015*) and crenactin single filament crystal structure (bottom) (*Izoré et al., 2014*), indicating that crenactin forms double filaments under the conditions used. (**C**) Comparison of cryoEM density maps obtained from double and single filament starting models. Left: crenactin double filament starting model with twist: 198.1° (equivalent to −161.9°) and rise: 25.6 Å applied during helical reconstruction. Right: crenactin single filament starting model with twist: 36.2° and rise: 51.3 Å applied. The two reconstructions converged to the same, double-stranded solution. (**D**) Part of the 3.8 Å cryoEM density map (resolution estimate determined through gold standard FSC 0.143 criterion, *Figure 1—figure supplement 1*). (**E**) Surface representation of crenactin double-stranded helical filament. See also *Video 1*. (**F**) Schematic showing the organisation of the arcade gene cluster, also showing similarities to proteins of known function (*Ettema et al., 2011*).

*Figure 1 continued on next page*

*Figure 1 continued*

The following figure supplement is available for figure 1:

**Figure supplement 1.** Fourier shell correlation (FSC) plot.

model (*Figure 1D*) of the crenactin double helical filament at near-atomic resolution (*Figure 1—figure supplement 1*, *Video 1*, Table S1 and Materials and methods).

## Crenactin double-helical filaments are exceptionally similar to F-actin

Crenactin forms right-handed, double-stranded, staggered filaments with a rise of 25.6 Å (half a subunits' length, hence staggered) between subunits and a twist of 198.1° (1-start, rotating between the two strands, equivalent to -161.9°) (*Figure 1E*). These values are very similar to the parameters of the eukaryotic F-actin filament, with a rise of 27.5 Å and a twist of 193.6° (equivalent to $-166.4°$, *Figure 2A*, *Video 2*, *Figure 2—figure supplement 1*) (*von der Ecken et al., 2015*). Architecture and helical parameters further add to the previously reported similarities in sequence and subunit structure (*Ettema et al., 2011*; *Izoré et al., 2014*; *Lindås et al., 2014*), making crenactin the closest F-actin homologue of any other actin-like filament investigated to date.

In F-actin, the double helix is stabilised via a so-called 'hydrophobic plug' (*Figure 2—figure supplement 2A*) (*Holmes et al., 1990*; *von der Ecken et al., 2015*), a loop of 10 amino acids (residues 263–272), between subdomains IIA and IIB, protruding into the inter-strand interface in F-actin filaments. In crenactin, this loop is longer and encompasses residues 292 to 326 (34 residues) (*Izoré et al., 2014*). All previous crenactin structures (two crystal-structures, PDB IDs 4CJ7, 4BQL, and one cryoEM reconstruction) (*Braun et al., 2015*; *Izoré et al., 2014*; *Lindås et al., 2014*) showed crenactin to be single-stranded and it was proposed that the position of the hydrophobic plug was incompatible with the formation of an F-actin-like double-stranded helix because of steric hindrance (*Braun et al., 2015*).

In our filament structure presented here, we observed a dramatic rearrangement of the hydrophobic plug, moving it upwards, towards subdomain IB, by as much as 21 Å (*Figure 2B*, bottom panel). In this new position, the loop interacts extensively with subunits of the opposite strand (*Figure 2—figure supplement 2B*), essentially the same function as the hydrophobic plug has in F-actin. As in F-actin, most of these lateral interactions are of hydrophilic nature in contrast to its name (*von der Ecken et al., 2015*). In addition to the hydrophobic plug, a hydrophilic interaction between subdomain IIB on one strand and subdomain IA on the opposite strand helps to keep the double filament architecture stable (*Figure 2—figure supplement 2C*). Based on the mostly hydrophilic nature of the inter-strand interactions, we believe it is possible that the single-stranded filaments imaged in previous studies were enabled by the high-salt concentrations used (*Braun et al., 2015*).

For longitudinal assembly, crenactin subunits within the same strand are held together via three main areas of interactions (*Figure 2C*, *Figure 2—figure supplement 1* and *Figure 2—figure supplement 2D*). Particularly significant is the well-conserved D-loop (DNase I binding loop) located within subdomain IB. Folded into a small alpha helix, it interacts with the previous subunit via a surface usually referred to as the 'hydrophobic groove' in actin (*Dominguez, 2004*).

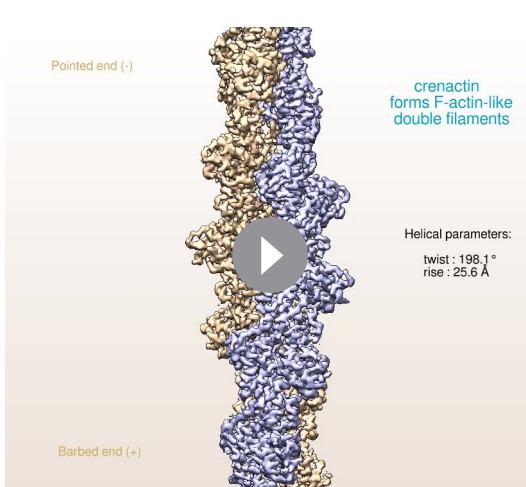

Pointed end (-)

crenactin forms F-actin-like double filaments

Helical parameters:

twist : 198.1 °
rise : 25.6 Å

Barbed end (+)

**Video 1.** Crenactin forms F-actin-like double filaments. The movie shows the experimental electrostatic potential density obtained from cryoEM and helical reconstruction and a ribbon representation of the refined atomic model of the filament.

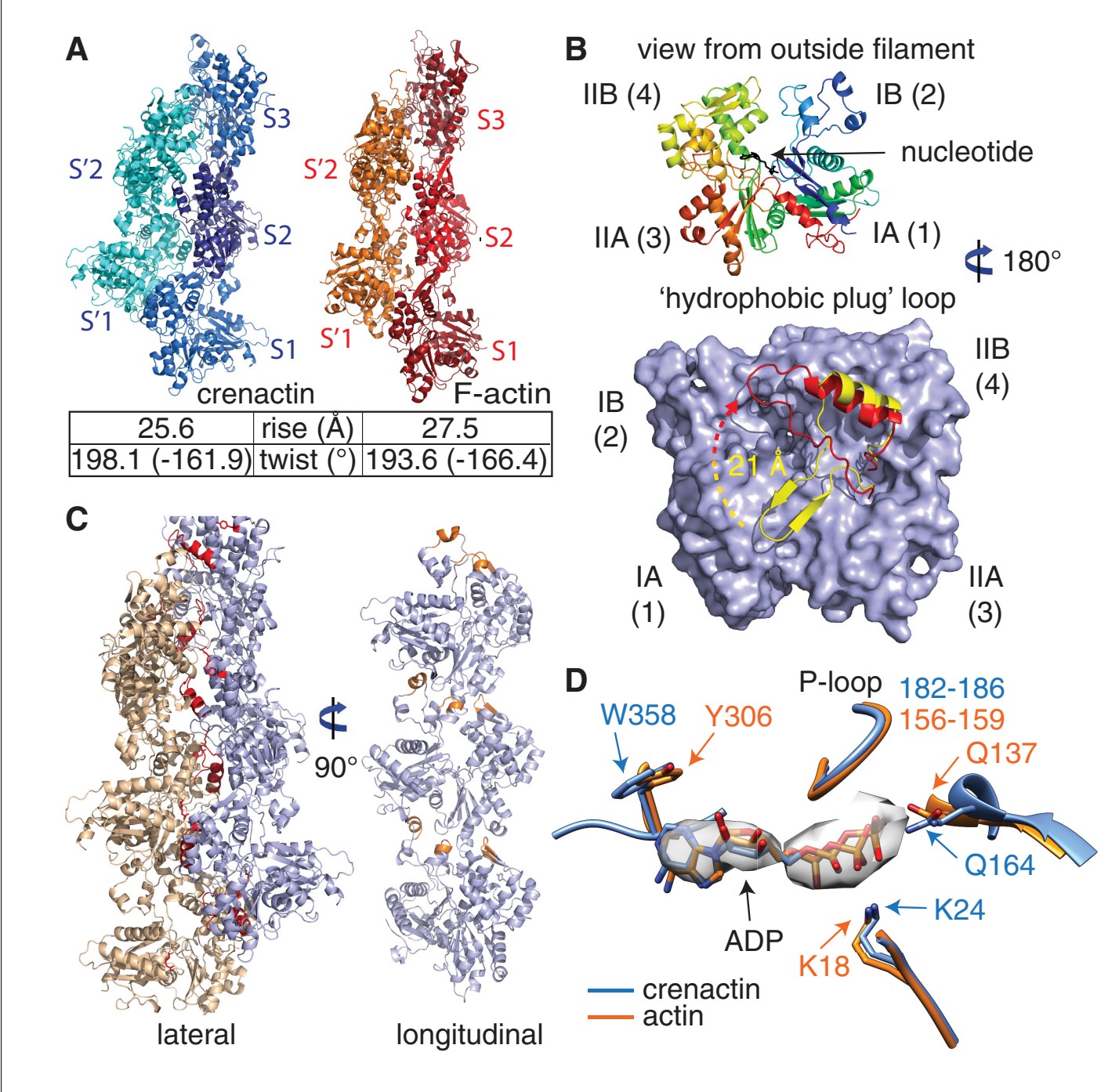

**Figure 2.** Crenactin filaments are exceptionally similar to eukaryotic F-actin. (**A**) Comparison of filament architectures between crenactin (blue) and actin (orange) (*von der Ecken et al., 2015*) showing the exceptional similarity that is also reflected in the helical parameters (bottom). F-actin model PDB ID: 3J8A (*von der Ecken et al., 2015*). See also *Video 2*. Subunits labelled S2 were superimposed for the comparison. (**B**) Top: cartoon plot of crenactin in the filament conformation, showing the common subdomain nomenclature used and the position of the nucleotide. View is from the outside of the double filament, with the 'hydrophobic plug' loop positioned in the back, inside the double filament. Bottom: crenactin's hydrophobic plug loop conformational change from the single-stranded filament form (yellow) to double stranded form (red). View from inside of filament, 180° rotated from top panel. (**C**) Lateral (red) and longitudinal (orange) interactions in crenactin double filaments. Longitudinal interactions are represented in a single crenactin strand for clarity. For a comparison to F-actin, please see *Figure 2—figure supplement 1*. (**D**) Comparison of the ATPase active site of crenactin (blue) and actin (orange). The cryoEM density for the ADP nucleotide is superimposed.

The following figure supplements are available for figure 2:

*Figure 2 continued on next page*

*Figure 2 continued*

**Figure supplement 1.** Comparison of longitudinal contacts between crenactin filaments and F-actin.

**Figure supplement 2.** Detailed lateral and longitudinal interactions within crenactin filaments.

Although the size of the hydrophobic plug is unique to crenactin, the longitudinal and lateral interactions between neighbouring subunits are similar to an unprecedented extent between crenactin and F-actin. The similarity in actin and crenactin helical parameters is striking, with only a difference of ~2 Å in rise and ~4.5° in twist (*Figure 2A*). Furthermore, the ATPase active site is also highly conserved with key residues, such as the nucleophilic water activator Q164 (Q137 in actin) (*Iwasa et al., 2008*), in the same place in both structures (*Figure 2D*). Because crenactin non-polymerising mutants (V339K and E340K) (*Izoré et al., 2014*) were resistant to crystallographic studies, we compared crenactin with one intra-strand contact (hydrophobic groove interacting with subdomain IB/D-loop) impaired by the presence of the arcadin-2 C-terminal peptide (see below) with a crenactin subunit from the double-helical filaments. As has been reported for many actins and actin-like proteins (*Fujii et al., 2010*; *Gayathri et al., 2012*; *van den Ent et al., 2014*), ATP binding, and more importantly polymerisation, induce inter-domain angle conformational changes, closing the groove between domains IB and IIB and removing a propeller twist of domains I and II against each other upon polymerisation. We also observed this change for crenactin. The rotation flattens the molecule in the polymer, removing the propeller twist between domains I and II (*Figure 2—figure supplement 2E*). This motion is conserved in both eukaryotic actin (*Fujii et al., 2010*; *von der Ecken et al., 2015*) and its prokaryotic homologues MreB (*van den Ent et al., 2014*) and ParM (*Gayathri et al., 2012*) and is most likely a pre-requisite of ATPase switching upon longitudinal polymerisation for the entire actin-like protein family.

## Crenactin interacts with arcadin-1 and -2 and arcadin-2 sequesters crenactin monomers

Since eukaryotic actin and its polymerisation are regulated through the action of a multitude of modulator proteins, we hypothesised that crenactin might interact with arcadins as they are encoded within the same arcade cluster (*Ettema et al., 2011*) (*Figure 1F*).

A sequence alignment of several arcadin-2s revealed that a small conserved C-terminal domain is separated from the core of the protein by a non-conserved, presumably unstructured linker (*Figure 3—figure supplement 1*). We investigated the effects of arcadin-2 on crenactin filament assembly by using 90° light scattering. Addition of ATP to crenactin resulted in an increase of scattering, most likely because of the formation of filaments (*Figure 3A*). Subsequent addition of arcadin-2 resulted in rapid depolymerisation of the polymers as scattering diminished. Addition of a C-terminally truncated version of arcadin-2 had no effect, suggesting that the conserved C-terminal α-helix of arcadin-2 was responsible for the depolymerisation of crenactin filaments. To test this, we performed the same experiment using a peptide spanning the last 17 amino acids of arcadin-2 (187–203). Indeed, the peptide triggered depolymerisation of the filaments at a similar rate as full-length arcadin-2 (*Figure 3A*). This effect was confirmed

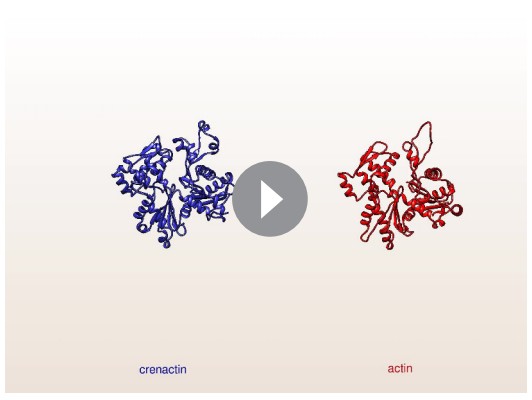

**Video 2.** Exceptional similarity between crenactin and actin. As was previously reported, the structure of crenactin subunits very closely resembles that of eukaryotic actin, including the 'hydrophobic plug' loop. With the cryoEM filament structure we show that this similarity extends to the filament architecture, with the two filament structures being exceptionally similar. A section with two and three subunits in each strand of the double helical filaments is shown.

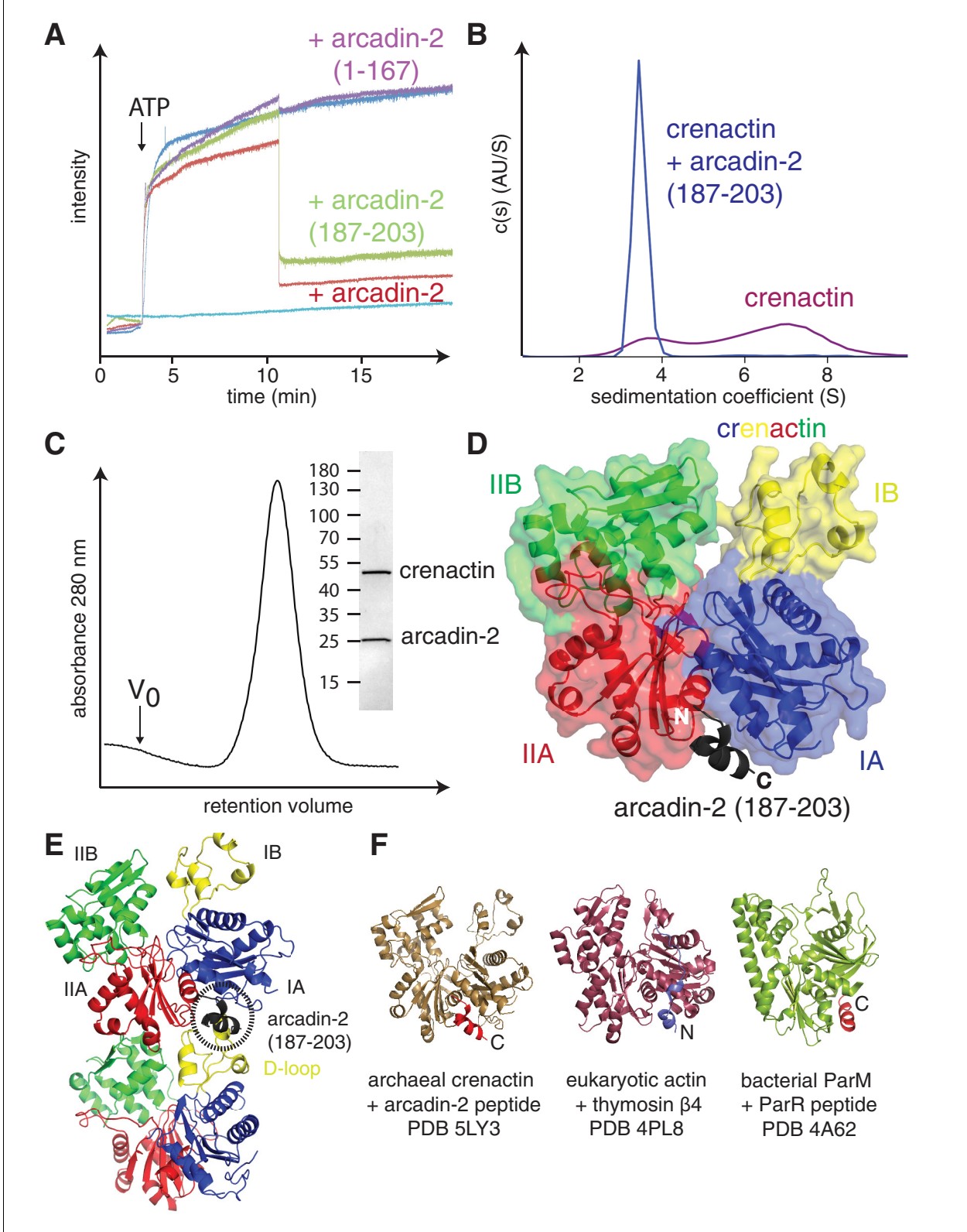

**Figure 3.** Crenactin interacts with arcadin-2. (**A**) 90° light scattering assay of crenactin polymerisation. Arrow indicates ATP addition. Crenactin polymerisation is shown in dark blue (positive control). Curves representing the depolymerisation of crenactin by addition of arcadin-2 and arcadin-2 C-terminal peptide (residues 187–203) are shown in red and green, respectively. A curve following the addition of arcadin2ΔC (residues 1–167, only) is shown in purple. Crenactin and arcadin-2 premixed before the experiment is shown with the light blue curve. (**B**) Analytical ultracentrifugation profile of

*Figure 3 continued on next page*

*Figure 3 continued*

crenactin and crenactin with arcadin-2 C-terminal peptide (residues 187–203), showing monomers only for the complex sample. (**C**) Size exclusion chromatography profile of the crenactin:arcadin-2 complex, with corresponding Coomassie-stained SDS-PAGE. (**D**) Ribbon/surface representation of crenactin:arcadin-2 peptide (residues 187–203) complex crystal structure at 1.6 Å resolution, showing the binding of arcadin-2 to the hydrophobic groove, where the D-loop binds in filaments of crenactin. (**E**) Ribbon representation of two subunits of crenactin in the filament. The localisation of the arcadin-2 C-terminal peptide (187–203) is shown in black. Note the clash between the presence of the arcadin-2 peptide and the polymer form of crenactin, especially the D-loop. (**F**) Ribbon representation of archaeal, eukaryotic and bacterial actins in complex with protein domains involved in the regulation of the filaments. PDB IDs: crenactin:arcadin-2 5LY3 (this work); actin:thymosin β4 4PL8 (*Xue et al., 2014*); ParM:ParR 4A62 (*Gayathri et al., 2012*). Note that the orientation of the thymosin peptide is reversed in comparison with arcadin-2 and ParR.

The following figure supplements are available for figure 3:

**Figure supplement 1.** Multiple sequence alignment of arcadin-2 sequences from a BLAST search showing a small C-terminal domain separated from the body of the protein by a non-conserved, presumably flexible linker.

**Figure supplement 2.** Biophysical and structural characterisation of the arcadin-2 : crenactin complex.

by analytical ultra centrifugation, showing that crenactin becomes monomeric upon addition of the C-terminal arcadin-2 peptide (*Figure 3B*). We conclude that crenactin polymerisation is controlled by arcadin-2 and that arcadin-2's C-terminal helix is essential for this activity.

Co-expression and subsequent purification of crenactin and arcadin-2 resulted in a tight 1:1 complex (*Figure 3C*). The dissociation constant of this complex was very low with a $K_d$ of 31 ± 4 nM (n = 4), as measured by SPR (Surface Plasmon Resonance, *Figure 3—figure supplement 2A and 2D*). To gain a better understanding of this interaction, how it triggers depolymerisation and how this might be related to F-actin depolymerisation, we solved the crystal structure of crenactin in complex with arcadin-2 C-terminal peptide. The structure was solved to 1.6 Å by X-ray crystallography (*Table 1*) providing detailed insights into crenactin, binding to ATP and, more importantly, showing how arcadin-2 induced depolymerisation of the filaments. The arcadin-2 peptide was located in the hydrophobic groove on crenactin, a cavity formed between subdomains IA and IIA (*Figure 3D*) (*Dominguez, 2004*). In the filament, this hydrophobic groove is occupied by the D-loop of the following subunit in a strand, forming one half of the longitudinal contact (*Figure 3E*). The nanomolar interaction between arcadin-2 peptide and the hydrophobic pocket is likely due to a tryptophan and other hydrophobic residues, making strong contacts as shown in *Figure 3—figure supplement 2E*. Additional SPR experiments showed that arcadin-2 C-terminal residues formed the only interacting domain with crenactin (*Figure 3—figure supplement 2A–D*). Given the high affinity of arcadin-2 for crenactin we propose that arcadin-2 disrupts crenactin filaments by competing with the D-loop for the hydrophobic groove.

Many of the plethora of eukaryotic actin-interacting proteins use the same hydrophobic groove as arcadin-2 on crenactin as a binding site, such as ADF/cofilin (*Paavilainen et al., 2008*), gelsolin (*McLaughlin et al., 1993*), thymosin β4 (*Irobi et al., 2004*), ciboulot (*Hertzog et al., 2004*) and the Wiskott–Aldrich Syndrome protein WH2 domain (WASP) (*Chereau et al., 2005*), amongst others (*Dominguez and Holmes, 2011*). It has therefore been termed a 'hot spot' for actin-binding proteins and there is also one such case in bacteria, the ParM:ParR interaction that involves binding of the ParR adaptor protein in ParM's hydrophobic groove (*Figure 3F*) (*Gayathri et al., 2012*). Proteins binding to the hydrophobic groove of actin facilitate a variety of functions depending on additional interactions and their affinity, but it seems to us that the high affinity of arcadin-2 and its strong bulk depolymerisation activity makes it a possible functional homologue of actin sequesters such as thymosin β4.

Our finding that arcadin-2 inhibits crenactin polymerisation in a manner similar to known actin sequesters provides further evidence for the parallel evolution of crenactin and eukaryotic actin since they have both maintained the function of the hydrophobic groove (*Bernander et al., 2011*; *Guy and Ettema, 2011*). Because of a lack of any detectable sequence similarity between arcadin-2 and actin sequesters, arcadin-2's mode of action might have arisen by convergent evolution.

The arcade cluster encodes three more proteins: arcadin-1, -3 and -4 (*Figure 1F*). Arcadin-3 is small and arcadin-4 is related by sequence to SMC-like proteins, especially Rad50, based on the

**Table 1.** Crystallography and cryoEM data.

| Statistics | | | |
|---|---|---|---|
| Sample | *Pyrobaculum calidifontis* crenactin: arcadin-2 peptide | *Pyrobaculum aerophilum* arcadin-1 | *Pyrobaculum calidifontis* crenactin |
| NCBI database ID | WP_011850310.1 WP_011850311.1 | NC_003364.1 | WP_011850310.1 |
| Constructs | crenactin 1-432, arcadin-2 187-203 | MGSSH$_6$SSGLVPRGSH-1-113 | 1-432 |
| **Method** **Data collection** | crystallography molecular replacement | crystallography SIR | cryoEM with helical reconstruction in RELION 2.0 |
| Beamline/microscope Wavelength / energy | Diamond I04-1 0.92819 Å | Diamond I04 0.97949 Å | FEI Polara / Falcon III 300 kV |
| **Crystal / helical parameters** | | | |
| Space / point group Cell (Å/°) | P2$_1$ 54.2, 70.9, 62.2, 104.21° | P6$_5$22 84.0, 84.0, 61.3, 90,90,120° | 1-start helical |
| Twist / rise | | | 198.1° (= −161.9°), 25.6 Å |
| **Data** | | | |
| Resolution (Å) | 1.6 | 2.0 | 3.8 |
| Completeness (%)* | 97.8 (93.5) | 100.0 (99.7) | |
| Multiplicity* | 3.3 (2.8) | 19.0 (15.1) | |
| (I) / °(I)* | 14.1 (1.7) | 26.6 (2.3) | |
| $R_{merge}$* | 0.037 (0.554) | 0.065 (1.348) | |
| $R_{pim}$* CC1/2 | 0.023 (0.378) 0.999 (0.897) | 0.015 (0.307) 1.00 (0.803) | |
| Images, pixel size Defocus range, dose Helical segments | | | 1474, 1.34 Å -0.8 - -3.0 μm, ~40 e/Å† 470396, 25 Å apart |
| **Refinement** | | | |
| R / $R_{free}$2† | 0.175 / 0.199 | 0.209 / 0.230 | 0.260 |
| Models | 2 chains crenactin: 4-430 arcadin-2: 188-203 ADP, 337 waters | 1 chain SH-1-32…72-113 37 waters | 6 chains refined in P1: 5-430, ADP, no waters |
| Bond length rmsd (Å) | 0.006 | 0.020 | 0.016 |
| Bond angle rmsd (°) | 0.860 | 2.12 | 1.453 |
| Favoured (%)‡ | 98.0 | 95.9 | 92.96 |
| Disallowed (%)‡ MOLPROBITY score | 0 100th percentile | 1.37 92nd percentile | 0.24 100th percentile |
| PDB/EMDB IDs | **5LY3** | **5LY5** | **5LY4, 4117** |

*Values in parentheses refer to the highest recorded resolution shell.

†5% of reflections were randomly selected before refinement.

‡Percentage of residues in the Ramachandran plot (PROCHECK 'most favoured' and 'additionally allowed' added together).

hinge regions. Little is known about arcadin-1. Our crystal structure of arcadin-1 shows it not to be related to any known eukaryotic actin binding proteins (*Figure 4A and B*) and also does not show obvious similarity to any other protein currently in the Protein Data Bank (PDB). Arcadin-1 forms tight dimers in the crystals (*Figure 4B*) and also oligomers (*Figure 4C and D*), with octamers being

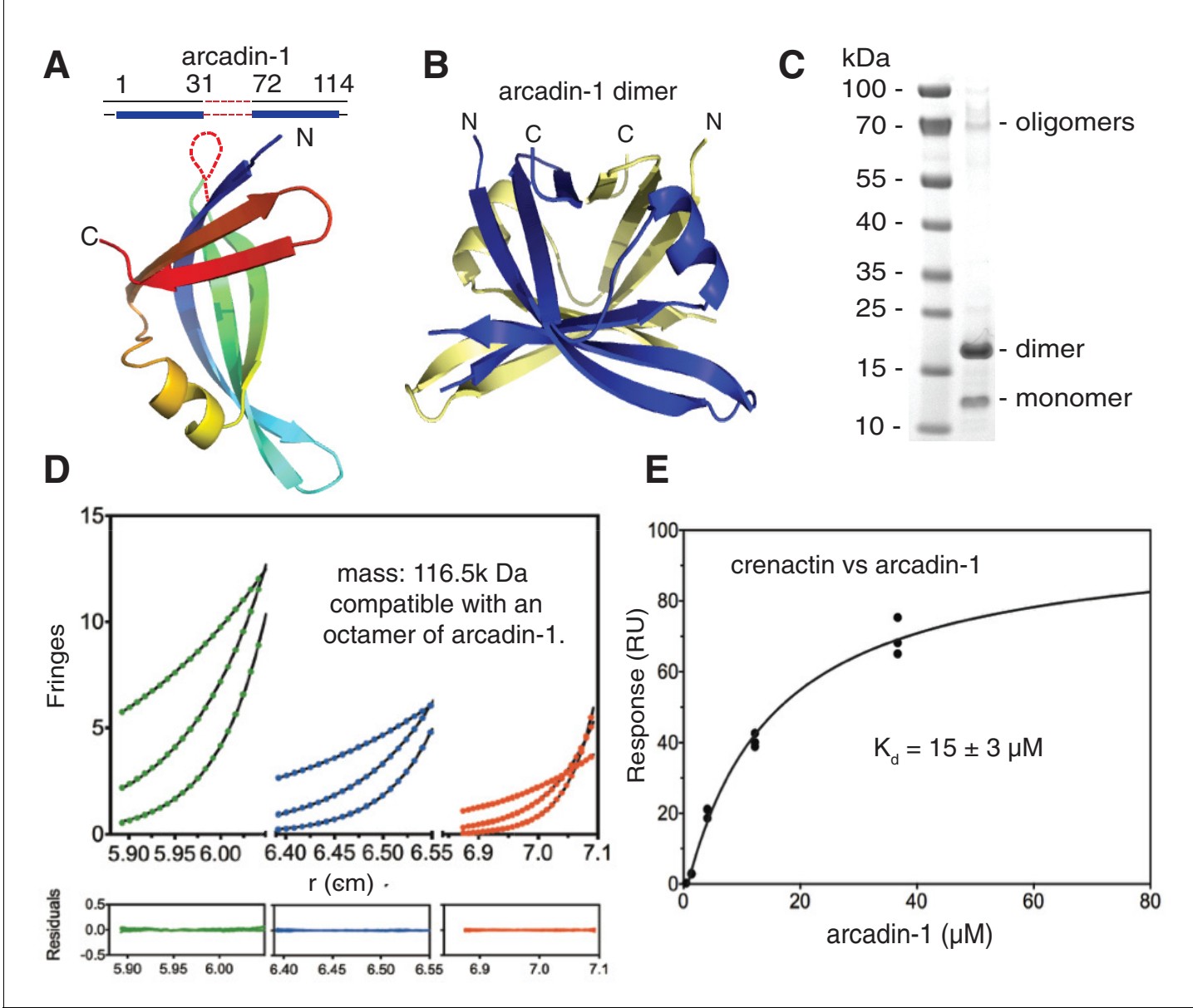

**Figure 4.** Crenactin interacts with arcadin-1. (A) Ribbon representation of the crystal structure of arcadin-1 at 2.0 Å resolution. Note that residues 32 to 71 are presumably disordered and missing from the structure. (B) Ribbon representation of the arcadin-1 dimer present in the crystal. The two subunits of the dimer have been coloured blue and pale yellow. (C) SDS-PAGE of arcadin-1. The protein appears mainly as a dimer, although monomers and oligomers can also be seen on the Coomassie-stained gel. (D) Sedimentation equilibrium analytical ultracentrifugation of arcadin-1. Sedimentation data for 199 µM (green closed circles) 99.5 µM (blue closed circles) and 49.8 µM (red closed circles) obtained at 7800, 11800, and 15,000 rpm were fitted to an idealised single-species model (solid lines). Every tenth data point is shown for clarity. The plots below show the residuals of the fits to the data. Analysis of multiple concentrations gave a molecular weight of 116,500 ± 273 Da, close to an octamer (monomer: 14,560 Da). (E) Surface plasmon resonance (SPR) of the interaction between arcadin-1 and crenactin. Equilibrium fitting for arcadin-1 association with crenactin gave a dissociation constant of $K_d = 15 \pm 3$ µM.

likely. A surface plasmon resonance assay revealed that arcadin-1 interacts moderately with crenactin ($K_d$ of $15 \pm 3$ µM, *Figure 4E*). However, arcadin-1 did not affect crenactin polymerisation as monitored by light scattering and EM. Further studies of the arcade cluster in vitro and in vivo will be required to learn about the interactions and functions of the proteins and the entire system with archaeal actin at its centre.

Our double helical filament structure of the crenarchaeal actin-like protein crenactin corrects previous reports of single crenactin strands (*Braun et al., 2015*) and firmly re-affirms the idea that Crenarchaea and eukaryotic cells share a common ancestor (*Guy and Ettema, 2011*). This is further supported by our finding that the arcade cluster contains other similarities to today's actin cytoskeleton as exemplified by arcadin-2's ability to depolymerise crenactin through interaction with the hydrophobic groove (*Dominguez, 2004*), most likely sequestering monomers with high affinity - a mode of action utilised by several eukaryotic actin modulators.

The recent discovery by metagenomics of Lokiarchaeota (*Spang et al., 2015*), being monophyletic with Eukaryotes, has unearthed actin homologues that are much closer to eukaryotic actin as judged by sequence identity than crenactin and we expect these to provide further evidence for how evolution progressed from a potential crenarchaeal ancestor to Eukaryotes, although it is clear from our work that primordial crenactin already provided an excellent template for the evolution and the origin of the eukaryotic actin cytoskeleton.

# Materials and methods

## Expression and purification of *Pyrobaculum calidifontis* crenactin

The codon-optimised gene encoding *Pyrobaculum calidifontis* crenactin (Genscript), database identifier WP_011850310.1, was subcloned into plasmid pOPIN-S (*Berrow et al., 2007*). The resulting construct encoded an N-terminal hexa-histidine SUMO-tag followed by crenactin. *E. coli* C41(DE3) (Lucigen, Middleton - Wisconsin) cells were transformed by electroporation with the pOPIN-S vector containing the crenactin insert and incubated overnight at 37°C on a agar plate supplemented with 50 µg/ml kanamycin. Cells were harvested from the plate and used to inoculate 120 ml of 2xTY media containing 50 µg/ml kanamycin. After reaching saturation, 120 ml were used to inoculate 12 L of TB (Terrific-Broth) media with kanamycin. The culture was first grown at 37°C until $OD_{600}$ reached 0.4, then for 1 hr at 18°C before protein expression was induced by the addition of 1 mM IPTG and continued for 16 hr. Cells were then pelleted and resuspended in 400 ml buffer A (50 mM Tris/HCl, 200 mM NaCl, 1 mM DTT, 10% glycerol (v/v), 10 mM imidazole, pH 8.0) supplemented with DNase I, RNase A (Sigma, St Louis - Missouri) and EDTA-free protease inhibitor tablets (Roche, Basel - Switzerland). Cells were lysed using a Constant Systems (UK) cell disruptor operating at 25 kPSI and the lysate was clarified by centrifugation at 180,000x g for 45 min The clarified lysate was incubated in the presence of Ni-NTA agarose beads (10 ml, Qiagen, Germany) at 4°C for 1 hr. Beads were subsequently washed extensively with buffer A, buffer A supplemented with 500 mM NaCl and again buffer A using a gravity column. Protein elution was achieved by tag cleavage for 3 hr at 4°C using purified SUMO protease SENP1 fused to GST at a protein: protease ratio of 1: 30. SENP1 was purified as previously described (*Izoré et al., 2014*). Crenactin co-eluted with a high molecular chaperone (GroEL) that was removed by the size exclusion chromatography step. In order to remove SUMO protease, the mixture was subsequently incubated with a small amount of glutathione-sepharose beads (GE Healthcare, Chicago, Illinois) for 30 min at 4°C. Following centrifugation to remove the resin, the protein solution was loaded onto a Sephacryl S300 16/60 size exclusion column (GE Healthcare) pre-equilibrated in buffer B (50 mM $NH_4HCO_3$, 50 mM NaCl). Fractions containing pure crenactin were concentrated using a Centriprep concentrator (30 kDa MWCO, Millipore) to 6–10 mg/ml and flash frozen in liquid nitrogen.

## Structure determination of crenactin bound to C-terminal arcadin-2 peptide

A peptide encompassing residues R187 to R203 of arcadin-2 (database reference identifier WP_011850311.1) was synthesised (Generon, U.K) and dissolved in water to a final concentration of 20 mM. Prior to setting up crystal trays, crenactin at 7 mg/ml was mixed with the arcadin-2 peptide at a molar ratio of 1 to 3. Many initial hits were obtained using our in-house nano-litre crystallisation facility (*Stock et al., 2005*). After optimisation, the best crystals were grown by vapour-diffusion in a drop composed of 100 nl of reservoir solution (0.31 M sodium acetate, 12.8% (w/v) PEG 4000, 0.1 M sodium acetate, pH 4.5) and 100 nl of protein solution. Crystals appeared in 1 day. Crystals were cryo-protected by passing them through a drop of reservoir supplemented with 30% (v/v) glycerol before flash freezing in liquid nitrogen. Datasets were collected at Diamond Light Source

(Harwell, UK) on beamline I04-1 on a Pilatus detector (Dectris, Switzerland). Data processing was performed using XDS (*Kabsch, 2010*) followed by merging in CCP4 (*Collaborative Computational Project, Number 4, 1994*). Phases were obtained by molecular replacement with PHASER (*McCoy et al., 2007*) using crenactin as a search model (PDB ID: 4CJ7_A, [*Izoré et al., 2014*]). Extra electron density was clearly visible and was manually fitted with an arcadin-2 peptide atomic model. Cycles of manual building were performed using COOT (*Emsley and Cowtan, 2004*) coupled with refinement by REFMAC and PHENIX (*Adams et al., 2010*; *Murshudov et al., 1997*).

## Expression and purification of *P. calidifontis* and *P. aerophilum* arcadin-1

The codon optimised *P. calidifontis* arcadin-1 gene (database identifier YP_001056517.1) was obtained as linear DNA from Integrated DNA Technology (IDT, Coralville - Iowa), whereas the *P. aerophilum* gene (NP_559897.1) was PCR amplified from genomic DNA. The *P. aerophilum* gene was cloned using the NdeI and BamH1 sites of plasmid pET15b, encoding an N-terminally hexa histidine-tagged protein fusion. The *P. calidifontis* gene was cloned between the NdeI and BamH1 sites of plasmid pHis17, resulting in a C-terminal hexa-histidine tag. Expression and purification of both proteins followed a similar protocol. *E. coli* C41(DE3) cells (Lucigen) for *P. calidifontis* arcadin-1 and Rosetta-II (Merck Millipore, Billerica - Massachusetts) for *P. aerophilum* were transformed with the respective plasmid and grown over night on agar plates supplemented with 50 µg/ml ampicillin. Cells were harvested and used to inoculate 6 litres of 2xTY media. Cells were grown to O.D.$_{600}$ 0.6 at 37°C and protein expression was then induced by the addition of 1 mM IPTG for 3 hr. Cells were pelleted and resuspended in 25 mM CHES, 350 mM NaCl, 5 mM imidazole, 10% glycerol (v/v), 1 mM DTT, pH 9.0, lysed using a Constant Systems cell disruptor operating at 25 kPSI, clarified by centrifugation and loaded onto a Ni-NTA affinity column (GE Healthcare) pre-equilibrated in the same buffer. Protein elution was performed by stepwise increases of imidazole. Fractions containing arcadin-1 were pooled and further purified using a Sephacryl S300 16/60 size exclusion column (GE Healthcare) pre-equilibrated in buffer composed of 25 mM HEPES, 100 mM NaCl, 2 mM MgCl$_2$, pH 7.5. Fractions containing pure arcadin-1 were concentrated to 15–20 mg/ml using a Centriprep concentrator (10 kDa MWCO, Millipore) and flash frozen in liquid nitrogen. All experiments except crystallography were performed using *P. calidifontis* arcadin-1 since only *P. aerophilum* arcadin-1 produced diffraction-quality crystals.

## Structure determination of arcadin-1 from *P. aerophilum*

Initial crystallisation hits were produced using our in-house nano-litre crystallisation facility (*Stock et al., 2005*). After optimisation, the best crystals were obtained in 200 nl drops composed of 100 nl of mother liquor (7.2% MPD (v/v), 14 mM MgCl$_2$, 50 mM sodium cacodylate pH 6.0) and 100 nl of protein concentrated to 16 mg/ml. Crystals were harvested, cryoprotected with 30% glycerol in reservoir solution and flash-frozen in liquid nitrogen. Phases for arcadin-1 were obtained by soaking crystals for 30 s in a solution made of the reservoir supplemented with 300 mM potassium iodide. Crystals were then cryoprotected with 30% glycerol and flash frozen in liquid nitrogen as before. A single wavelength anomalous dispersion experiment (SAD λ= 1.5419 Å) was performed in-house using a FrE+ (Rigaku, Tokyo - Japan) rotating anode generator coupled to a mar345DTB image plate detector. Data were processed to 2.9 Å using XDS (*Kabsch, 2010*) and merged in CCP4 (*Collaborative Computational Project, Number 4, 1994*). AutoSHARP (*Vonrhein et al., 2007*) found 1 iodide site and produced an initial electron density map and model. A higher resolution dataset to 2 Å, collected at Diamond Light Source (Harwell, UK) on beamline I04 was solved using the initial model from the iodide dataset as a search model for molecular replacement using PHASER (*McCoy et al., 2007*). Cycles of manual building were performed using COOT (*Emsley and Cowtan, 2004*) cycled with refinement by REFMAC and PHENIX (*Adams et al., 2010*; *Murshudov et al., 1997*).

## Expression and purification of *P. calidifontis* arcadin-2 and arcadin-2 C-terminal truncation mutant (arcadin-2ΔC)

The gene encoding *P. calidifontis* arcadin-2 (database identifier WP_011850311.1) was obtained codon-optimised from Integrated DNA Technology (IDT). Following PCR amplification, the gene was sub-cloned into plasmid pHis17 using NdeI and BamH1 restriction sites. The resulting construct was transformed into *E. coli* C41(DE3) cells (Lucigen) by electroporation to produce untagged, native

protein. Following an over night pre-culture, 6 litres of 2xTY media, supplemented with 50 µg/ml ampicillin were inoculated. The culture was first grown at 37°C until $OD_{600}$ reached 0.4, then for 1 hr at 20°C before protein expression was induced by the addition of 1 mM IPTG and continued for 16 hr. Cells were collected by centrifugation and lysed in buffer A (50 mM HEPES, 400 mM NaCl, 1 mM DTT, 1 mM EDTA, pH 7.5) using a Constant Systems cell disruptor operating at 25 kPSI. After a first centrifugation (180,000 x g for 45 min), the clarified lysate was heated to 60°C for 20 min in a water bath to remove heat-labile *E. coli* proteins and centrifuged again. The soluble, heat-resistant proteins from the supernatant were subsequently concentrated by ammonium sulphate precipitation at room temperature. The precipitate was resuspended in buffer A and loaded onto a Sephacryl S300 16/60 size exclusion column (GE Healthcare) pre-equilibrated in buffer B (25 mM HEPES, 400 mM NaCl, 1 mM EDTA, 1 mM DTT, pH 7.0). As judged by SDS-PAGE, fractions containing pure arcadin-2 were concentrated using a Centriprep concentrator (10 kDa MWCO, Millipore) to 4 mg/ml and flash frozen in liquid nitrogen. The arcadin-2 expressing pHis17 plasmid was used as a template to add a stop codon after residue E167 using Q5 site directed mutagenesis (New England Biolabs, Ipswich - Massachusetts). This led to a C-terminally truncated version of arcadin-2 spanning residues 1–167, only (arcadin-2ΔC). The purification of arcadin-2ΔC followed the same initial steps as the full-length protein; however, the size exclusion buffer was 25 mM Tris/HCl, 100 mM NaCl, 1 mM DTT, pH 8.0. Fractions containing arcadin-2ΔC were pooled and loaded onto a MonoQ 5/50 GL anion exchange column (GE-Healthcare), pre-equilibrated in buffer B (25 mM Tris/HCl, 1 mM DTT). Elution of the protein was achieved by a gradient to 1 M NaCl in buffer B. Fractions containing pure arcadin-2ΔC were pooled, concentrated using a Centriprep concentrator (10 kDa MWCO, Millipore) and flash frozen in liquid nitrogen.

## Crenactin polymerisation

For cryo-electron microscopy (cryoEM), crenactin was diluted to a final concentration of between 0.5 and 1 mg/ml in 50 mM $NH_4HCO_3$, 20 mM KCl (no pH adjustment) and polymerisation was induced by the addition of 2 mM ATP and 4 mM $MgCl_2$ for 30 min on ice. For 90° light scattering, 3 µM of crenactin were polymerised in 50 mM $NH_4HCO_3$, 50 mM NaCl, 8% (w/v) PEG 8000, 2 mM ATP and 4 mM $MgCl_2$ at room temperature.

## CryoEM data collection and structure determination

After polymerisation on ice, 3 µl of sample were pipetted onto a freshly glow-discharged Quantifoil Cu R2/2 200 mesh grid and plunge frozen into liquid ethane using a Vitrobot Mark III (FEI, Hillsboro - Oregon). The Vitrobot chamber temperature was set to 4°C and humidity to 100%. Micrographs of crenactin filaments were collected with an FEI Tecnai G2 Polara microscope operating at 300 kV. Data were acquired on a Falcon III direct electron detector protoype at a calibrated pixel size of 1.34 Å and a total dose of 40 e-/A°$^2$ using the automated acquisition software EPU (FEI). Images were collected at 0.8 to 3.0 µm underfocus and dose-fractionated into 46 movie frames (30 fps). All image processing and helical reconstructions were done using RELION 2.0 (*Scheres, 2012*) that implements single particle real-space helical reconstruction IHRSR (*Egelman, 2007*). Briefly, this implementation performs single-particle-like processing of helical assemblies in an empirical Bayesian framework, where a marginalised likelihood function is complemented with a prior on the reconstruction that effectively dampens high spatial-frequency terms in the absence of experimental data. A total of 1474 micrographs were collected and drift-corrected using MOTIONCORR (*Li et al., 2013*). The contrast transfer function (CTF) was estimated by GCTF (*Zhang, 2016*). Filament segments were first manually picked on several micrographs, extracted as square boxes of 280 pixels and classified using reference free 2D classification. A subset of six 2D class averages representative of the different filament orientations were low-pass filtered to 20 Å and used as references to automatically pick the entire dataset with overlapping helical segments 25 Å apart in 280 pixel boxes. Autopicking accuracy was increased by identifying filaments and their directions and also their bending in RELION. Helical segments were split into two half datasets for gold standard FSC determination by keeping segments from each filament in one of the two half sets, avoiding over-fitting through comparing the same parts of images because of the picking of overlapping segments. After removing bad segments, 470,396 segments remained and were entered into 3D auto-refinement using 30 Å low-pass filtered initial models, generated from the approximated symmetry of crenactin

double filaments or single filaments and crenactin's monomer structure (*Izoré et al., 2014*). Beam-induced drift was subsequently corrected for per particle and frame-based dose weighting was applied (*Scheres, 2014*), leading to particles with increased signal to noise ratio since the final reconstruction and postprocessing produced the highest resolution map at 3.8 Å, as assessed by the gold standard FSC procedure implemented in RELION (0.143 FSC criterion, *Figure 1—figure supplement 1*) (*Rosenthal and Henderson, 2003*). Postprocessing used a mask covering the central 30% of the map, surrounded with an eight-pixel wide soft raised cosine edge. The FSC procedure was modified such that the two half sets contained particles from complete filaments, each, avoiding over-fitting through the use of very similar particles in the two half sets generated from overlapping filament segments. The electrostatic potential density map was visualised in UCSF Chimera (*Goddard et al., 2007*) and a model for the double filament consisting of six monomers in two strands (three each) was built by placing crenactin monomers (PDB ID 4CJ7) (*Izoré et al., 2014*) in the density map by molecular replacement with PHASER using phased translation functions (*McCoy et al., 2007*) and the model was manually adjusted and corrected with MAIN (*Turk, 2013*). Refinement of the model was carried out against density cut out around 6 central monomers as implemented in REFMAC (*Brown et al., 2015*). REFMAC, PHENIX.refine in real-space mode (*Adams et al., 2010*) and manual building in MAIN (*Turk, 2013*) were cycled until the best fit of the model into the original density map was achieved. For statistics of refinement please refer to *Table 1*. *Figure 1—figure supplement 1* also shows the FSC curve (red) of the refined atomic model against the post processed map cut around the six monomers of the model and not low-pass filtered. An FSC criterion of 0.5 (*Rosenthal and Henderson, 2003*) yielded the same resolution of 3.8 Å, together with the reciprocal space R-factor (*Table 1*) demonstrating the overall correctness of the model. All figures were prepared using PyMOL and Chimera (*Goddard et al., 2007*).

## 90° Light scattering polymerisation assays

Light scattering experiments were carried out on a Cary Eclipse spectrometer (Varian, Palo Alto - California) in a 100 μl quartz cuvette. Kinetics were recorded at 25°C over 25 min, with excitation and emission wavelengths of 360 nm and a 5 nm slit width. In all experiments 2 mM ATP and 4 mM $MgCl_2$ were added 2 min after the beginning of the experiment. After 10 min, 4 μM of arcadin-2 (or otherwise stated) were added and the experiment carried on for 15 additional minutes. All experiments were measured as triplicates.

## Analytical ultracentrifugation

Equilibrium sedimentation experiments for arcadin-1 were performed on an Optima XL-I analytical ultracentrifuge (Beckman, Brea - California) using An50Ti rotors. Sample volumes of 110 μL with protein concentrations of 49.8, 99.5 and 199 μM were loaded in 12 mm 6-sector cells and centrifuged at 7800, 11800, and 15,000 rpm until equilibrium was reached at 20°C. At each speed, comparison of several scans was used to judge whether or not equilibrium had been reached. Buffer conditions were in 25 mM Tris-HCl, pH 8.0, 100 mM NaCl, 1 mM EDTA. Data were processed and analysed using UltraSpin software (http://www.mrc-lmb.cam.ac.uk/dbv/ultraspin2/) and SEDPHAT (*Schuck, 2003*). Velocity sedimentation of samples of 2 μM crenactin in the absence and presence of 2 μM arcadin-2 C-terminal peptide was carried out at 50,000 rpm at 20°C in PBS using 12 mm double sector cells in an An50Ti rotor. The sedimentation coefficient distribution function, c(s), was analysed using the SEDFIT program, version 14.0 (*Schuck, 2003*). The partial-specific volumes (v-bar), solvent density and viscosity were calculated using SEDNTERP (personal communication, Thomas Laue, University of New Hampshire, USA). Data were plotted with the program GUSSI (*Brautigam, 2015*).

## Surface plasmon resonance (SPR)

SPR was performed using a Biacore T200 instrument using CM5-sensor chips (GE Healthcare). Both reference control and analyte channels were equilibrated in PBS-0.005% (v/v) Tween 20 at 20°C. Crenactin was immobilised onto the chip surface through amide coupling using the supplied kit (GE Healthcare) to reach an RU value of ~2300 for arcadin-1 experiments, ~700 for arcadin-2 and arcadin-2ΔC, and ~3000 for arcadin-2 C-terminal peptide experiments. SPR runs were performed in triplicate with analytes injected for 120 s followed by a 600 s dissociation in 1:3 dilution series with initial

concentrations of arcadin-1 from 36.7 µM or in 1:2 dilution series with initial concentrations of arcadin-2 and arcadin-2 ΔC from 2 µM and of C-terminal peptide from 750 nM. The surface was regenerated with 200 mM sodium carbonate, pH 11.0 for 120 s.

After reference and buffer signal correction, sensogram data were fitted using KaleidaGraph (Synergy Software) and Prism (GraphPad Software Inc). For Arcadin-1, the equilibrium response ($R_{eq}$) data were fitted using a single site interaction model to determine $K_d$:

$$R_{eq} = \left( \frac{CR_{\max}}{C + K_d} \right) \tag{1}$$

where C is the analyte concentration and $R_{\max}$ is the maximum response at saturation.

For Arcadin-2 and C-peptide kinetics, the rate constants of dissociation were measured by fitting dissociation data at time t ($R_{dissoc}$) using a single or double-exponential function:

$$R_{dissoc} = R_o \exp^{-\left(k_{off}t\right)} + RI + Dt \tag{2}$$

$$R_{dissoc} = R_{o1} \exp^{-\left(k_{off1}t\right)} + R_{o2} \exp^{-\left(k_{off2}t\right)} + RI \tag{3}$$

where $k_{off}$ is the dissociation rate constant, $R_o$ is maximum change in resonance each phase, RI is the bulk resonance change and D is a linear drift term. The rate constants of association were obtained by fitting the observed change in resonance signal ($R_{assoc}$) at time t using the following equation:

$$R_{assoc} = \left( \frac{k_{on}CR_{max}}{k_{on}C + k_{off}} \right) \left[ 1 - \exp^{-\left(k_{on}C + k_{off}\right)t} \right] + RI + Dt \tag{4}$$

$$R_{assoc} = \left( \frac{k_{on1}CR_{max1}}{k_{on1}C + k_{off1}} \right) \left[ 1 - \exp^{-\left(k_{on1}C + k_{off1}\right)t} \right] + \left( \frac{k_{on2}CR_{max2}}{k_{on2}C + k_{off2}} \right) \left[ 1 - \exp^{-\left(k_{on2}C + k_{off2}\right)t} \right] + RI \tag{5}$$

where $k_{on}$ is the association rate constant, C is the analyte concentration and $R_{\max}$ is the maximum change in resonance. The affinity for the interactions were calculated from the ratios of the microscopic rate constants:

$$K_d = \frac{k_{off}}{k_{on}} \tag{6}$$

The observed rate constant of association for C-peptide and PCC was obtained from fits to a single exponential function at each concentration:

$$R_t = R_o \left( 1 - \exp^{-\left(k_{obs}t\right)} \right) + RI \tag{7}$$

where $k_{obs}$ is the observed association rate constant. Data were fitted to a pseudo-first order association:

$$k_{obs} = k_{on}C + k_{off} \tag{8}$$

where C is the total concentration of C-peptide.

## Acknowledgements

This work was funded by the Medical Research Council (U105184326 to JL) and the Wellcome Trust (095514/Z/11/Z to JL). TI was the recipient of an EMBO Long Term Fellowship (ALTF 1379-2011).

## Additional information

### Funding

| Funder | Grant reference number | Author |
| --- | --- | --- |
| Medical Research Council | U105184326 | Danguole Kureisaite-Ciziene Stephen H McLaughlin Jan Löwe |
| Wellcome | 095514/Z/11/Z | Thierry Izoré |

| | | Jan Löwe |
|---|---|---|
| European Molecular Biology Organization | ALTF 1379-2011 | Thierry Izoré |

The funders had no role in study design, data collection and interpretation, or the decision to submit the work for publication.

## Author contributions

TI, Conception and design, Acquisition of data, Analysis and interpretation of data, Drafting or revising the article; DK-C, SHM, Acquisition of data, Analysis and interpretation of data; JL, Conception and design, Analysis and interpretation of data, Drafting or revising the article

## Author ORCIDs

Stephen H McLaughlin, http://orcid.org/0000-0001-9135-6253
Jan Löwe, http://orcid.org/0000-0002-5218-6615

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
