## [Decision Letter]

Thank you for submitting your article "Crenactin forms actin-like double helical filaments regulated by arcadin-2" for consideration by *eLife*. Your article has been reviewed by three peer reviewers, and the evaluation has been overseen by Werner Kühlbrandt as the Reviewing Editor and John Kuriyan as the Senior Editor. The following individual involved in review of your submission has agreed to reveal his identity: Robert Robinson (Reviewer #2).

The reviewers have discussed the reviews with one another and the Reviewing Editor has drafted this decision to help you prepare a revised submission.

All three reviewers agree that this is important work, that the paper is well-written and illustrated and suitable for publication in *eLife*. Nevertheless, they raise a number of concerns, which need to be addressed in a revised manuscript, as detailed below.

Summary:

This manuscript describes the cryoEM structure of a filament formed from crenactin, its regulation by arcadin-2, the co-crystal structure of crenactin/arcadin-2, and the crystal structure of arcadin-1. Together these data provide a fascinating insight into the structure and regulation of this filament system from Crenarchaea, which closely resembles F-actin. The structure is remarkable, and has increased interest due to a previous report that detailed a single-stranded architecture for the crenactin filament. The manuscript will find wide interest in the fields of evolution and filament architectures.

Essential revisions:

1) Since crenactin is as much of a modern protein as actin, the term "living fossil" should be used with caution. It is possible, but not certain, that crenactin has stayed closer to the common ancestor than actin itself. Indeed, the opposite could be argued from Figure 1 in Molecular Microbiology (2011) 80: 1052-1061.

2) The authors demonstrate that arcadin-2 controls the polymerization of crenactin by binding with its C-terminal end of to a hydrophobic groove of crenactin with nanomolar affinity. Given that several actin de-polymerization agents target this groove, they conclude that this protein could be a "functional homolog" to the eukaryotic proteins. They further propose that this is in line with the idea of the evolutionary links. Since they do not show that arcadin-2 has a similar fold to any actin regulator nor find any sequence similarity, arcadin-2 is not a sequence homologue of any actin-binding protein, and therefore does not have a common origin. The lack of sequence homology actually does not support the crenarcheal origin theory or parallel evolution. On the contrary, it could be seen as a case of convergent evolution.

3) The difference between the double-stranded helix and the single stranded helix shown by others is attributed to differences in salt conditions. However, very high salt conditions (close to 1M) are routinely used as a step in actin purification to remove binding proteins from F-actin while the filament stays intact. Since the intra- and interfilament interactions are similar in crenactin and actin, high salt concentrations should not affect the filament structure. If the salt concentration does have such a strong effect on the crenactin filaments, this is an important finding that should be buttressed, e.g. by a structure of the high-salt filaments in negative stain.

4) The method used for helical reconstruction is not entirely clear. In the absence of a paper describing helical refinement in Relion, the protocol should be explained at least briefly in the Methods. In particular, the authors segmented their filaments with a distance of 25 Å. Normally, this distance is chosen to be longer that the helix rise. However, the helix rise in this case is 25.8 Å, which raises the question of how the alignment of segments is handled by Relion. This is even more surprising for the reconstruction of a single strand, since the rise in this case is more than twice the distance that was used to segment the filaments. It is also unusual that the helical rotation is ~198˚, but by definition it has to be between 180˚ and -180˚. Are the numbers wrong or does the definition of helical rotation in Relion differ from the usual convention, and if so, why? Similarly, the actin twist is given as 193.6º, different from the commonly used ~167º. This may be due to a different definition of the coordinate system, in which case it should be explained.

5) Although the interactions are hydrophilic, it is not a good idea to rename the hydrophobic plug into F-loop. The actin field is quite mature and the term hydrophobic plug has been used for 20 years or more.

6) Subsection “Crenactin forms double-helical filaments”, first paragraph: The authors seem to suggest that the growth conditions for this organism control the salt/buffer conditions within the cell. Do they have a reference for this? Alternatively, a reference for the physiological salt concentrations within the organism should be given. Otherwise, the claim that 20 mM KCl is more physiological than 500 mM KCl appears to be speculation. Most organisms have KCl levels in the 100-350 mM range.

7) Irrespective of the KCl concentration within the organism, there is no doubt that the filament structure presented in this manuscript is a physiological filament. However, there are no data in the manuscript that address whether the previously published single-stranded filament is, or is not, a physiological filament. Do the authors rule out that there could be two physiological architectures? The authors should include some analysis to compare the protofilaments. Given the very different twists, it is not obvious how single-stranded filaments (published before) would form from the salt-induced dissociation of this double-stranded filament – although this is difficult to judge without the coordinates.

8) Introduction, second paragraph: The statement that the MreB structure is the most divergent from F-actin structure is incorrect. There is plenty of evidence that ParMs have the most varied architectures.

9) Figure 3—figure supplement 2 – These two figure panels are very important. They should be moved into the main paper. The direction of the sequestering helices in each case should be noted – they appear to vary.

---

## [Author Response]

*[…]*

*Essential revisions:*

*1) Since crenactin is as much of a modern protein as actin, the term "living fossil" should be used with caution. It is possible, but not certain, that crenactin has stayed closer to the common ancestor than actin itself. Indeed, the opposite could be argued from Figure 1 in Molecular Microbiology (2011) 80: 1052-1061.*

Many thanks for highlighting this issue. We used the term too loosely and agree it should not be used here as it implies that crenactin is closer to the common ancestor than F-actin. Modified the text to: "Hence, it has been proposed that crenactin filaments share a common ancestor with F-actin."

*2) The authors demonstrate that arcadin-2 controls the polymerization of crenactin by binding with its C-terminal end of to a hydrophobic groove of crenactin with nanomolar affinity. Given that several actin de-polymerization agents target this groove, they conclude that this protein could be a "functional homolog" to the eukaryotic proteins. They further propose that this is in line with the idea of the evolutionary links. Since they do not show that arcadin-2 has a similar fold to any actin regulator nor find any sequence similarity, arcadin-2 is not a sequence homologue of any actin-binding protein, and therefore does not have a common origin. The lack of sequence homology actually does not support the crenarcheal origin theory or parallel evolution. On the contrary, it could be seen as a case of convergent evolution.*

Correct – we did not identify any significant sequence homology between arcadin-2 and any other proteins. We were not trying to demonstrate that arcadin-2 is a sequence homologue of a eukaryotic actin-binding protein. We found that the mechanism of action of G-actin sequesters is similar to the mechanism we deciphered. Even if the sequence of arcadin-2 is divergent from known actin-binding proteins, it seems that the mechanism itself (crenactin filament inhibition) is homologous to the one used in eukaryotes and this is what we believe does support the theory of crenarchaeal origin as it makes actin and crenactin more related. It does seem more like convergent evolution on arcadin-2's side and we agree that this point was not made. The Discussion has been modified to include both points: "Our finding that arcadin-2 shares functional and structural similarities with actin sequesters provides further evidence for the parallel evolution of the crenarchaeal and eukaryotic cytoskeletons (Bernander et al., 2011; Guy and Ettema, 2011)." has been changed to: "Our finding that arcadin-2 inhibits crenactin polymerisation in a manner similar to known actin sequesters provides further evidence for the parallel evolution of crenactin and eukaryotic actin since they have both maintained the function of the hydrophobic groove (Bernander et al., 2011; Guy and Ettema, 2011). Because of a lack of any detectable sequence similarity between arcadin-2 and actin sequesters, arcadin-2's mode of action might have arisen by convergent evolution."

*3) The difference between the double-stranded helix and the single stranded helix shown by others is attributed to differences in salt conditions. However, very high salt conditions (close to 1M) are routinely used as a step in actin purification to remove binding proteins from F-actin while the filament stays intact. Since the intra- and interfilament interactions are similar in crenactin and actin, high salt concentrations should not affect the filament structure. If the salt concentration does have such a strong effect on the crenactin filaments, this is an important finding that should be buttressed, e.g. by a structure of the high-salt filaments in negative stain.*

Every actin-like protein is different and even single amino acid changes can have drastic effects on polymerisation conditions. Actin and crenactin share similar architectures at the filament level. However, many chemical differences exist at the monomer level. One of the most prominent differences is the hydrophobic plug as mentioned in the text and shown in figures, a feature that dominates inter-strand binding in crenactin and that is much smaller in actin. Even if the interactions are of similar nature, we suggest it is probably inappropriate to compare the polymerisation conditions of these two different proteins since their primary sequences differ significantly (only about 20% identity!). As an example, MreB, an endogenous bacterial actin-like protein, requires very different polymerisation conditions depending on the source organism, despite an extremely conserved 3D structure. MreB from *C. crescentus* does not polymerise at all if salt concentrations above 50 mM are used, whereas other MreB proteins show a different behaviour (Van den Ent 2014 *eLife*, Salje 2011 MolCELL, Van den Ent 2001 Nature). We cannot be sure entirely that it is the high salt condition that causes the different filament architectures, but the salt is the most striking difference between the conditions used.

Finally, it is suggested to provide a negative stain structure of the high-salt filaments. This has partially been done in our previous paper (Izoré 2014, FEBS Lett) where it is shown that under high salt conditions that crenactin most probably forms single stranded filaments (2D averaging). This has later been confirmed by a 18 Å cryoEM structure of the single-stranded filament (Braun 2015 PNAS) using the same conditions so we would like to suggest it is not needed here as it would be a duplication.

*4) The method used for helical reconstruction is not entirely clear. In the absence of a paper describing helical refinement in Relion, the protocol should be explained at least briefly in the Methods. In particular, the authors segmented their filaments with a distance of 25 Å. Normally, this distance is chosen to be longer that the helix rise. However, the helix rise in this case is 25.8 Å, which raises the question of how the alignment of segments is handled by Relion. This is even more surprising for the reconstruction of a single strand, since the rise in this case is more than twice the distance that was used to segment the filaments. It is also unusual that the helical rotation is ~198˚, but by definition it has to be between 180˚ and -180˚. Are the numbers wrong or does the definition of helical rotation in Relion differ from the usual convention, and if so, why? Similarly, the actin twist is given as 193.6º, different from the commonly used ~167º. This may be due to a different definition of the coordinate system, in which case it should be explained.*

Apologies, the manuscript describing helical reconstruction in Relion 2.0 is only now being readied for publication (Shaoda He & Sjors Scheres, in preparation). The Relion modifications essentially add a symmetry optimisation step (in real space) and a symmetrisation step (in reciprocal and/or real space) to the Refine3D procedure that has been described by Sjors previously (Scheres SHW, JSB 2012). Additionally, the procedure keeps segments from individual filaments in the two half datasets for FSC calculations separate, not to introduce bias and adds a powerful automatic picking procedure that tracks along filaments. The Methods section has been slightly amended to explain a bit more but the quality of the map and the agreement with the crenactin crystal structure (and the visible differences such as the hydrophobic plug conformation) speak for themselves we believe. The Relion additions will be published properly and it is not uncommon to publish structures with unpublished procedures (for example Phenix and CCP4 are updated/amended much more often than the literature).

The segments were picked 25 Å / 50 Å apart but are 375 Å (280 pixels x 1.34 Å) long. So they overlap a lot, one of the principles that makes IHRSR work despite the monomers being very small for cryoEM.

Relion does not impose any restrictions on the range of possible twist angles. Hence, we prefer to provide positive values for the twists when the resulting structures are right-handed. A twist of +198° is of course equivalent to -162°. Same for F-actin: +193 is equivalent to -167. To avoid confusion, both numbers are now listed everywhere. Most (all?) programs use the same direction of the rotation (but often the sign is omitted from tables and manuscripts).

*5) Although the interactions are hydrophilic, it is not a good idea to rename the hydrophobic plug into F-loop. The actin field is quite mature and the term hydrophobic plug has been used for 20 years or more.*

'F-loop' removed everywhere.

*6) Subsection “Crenactin forms double-helical filaments”, first paragraph: The authors seem to suggest that the growth conditions for this organism control the salt/buffer conditions within the cell. Do they have a reference for this? Alternatively, a reference for the physiological salt concentrations within the organism should be given. Otherwise, the claim that 20 mM KCl is more physiological than 500 mM KCl appears to be speculation. Most organisms have KCl levels in the 100-350 mM range.*

Unfortunately, we are not aware of any data indicating the intracellular salt concentration within the organism. The only available indication is the media the organism likes to grow in that is very dilute: tryptone 10 g/L, yeast extract 1 g/l, Na_2_S_2_O_3_ x 5H_2_O 3 g/L. Although the organism will maintain different conditions on the inside, we believe it is unlikely to be in the range of more than 250-350 mM KCl as indicated. We have weakened this argument in the text accordingly: "Previous studies on the architecture of crenactin filaments were performed under high salt concentrations (> 0.5 M KCl) (Braun et al., 2015; Izoré et al., 2014) that might not be entirely justified given *Pyrobaculum calidifontis'* environmental and laboratory growth conditions (Amo et al., 2002), although the intracellular osmolarity is currently not known. To exclude the possibility that such high salt concentration might have altered filament architecture, we carried out experiments in low-salt buffer (50 mM ammonium carbonate, 20 mM KCl, see Materials and methods)."

*7) Irrespective of the KCl concentration within the organism, there is no doubt that the filament structure presented in this manuscript is a physiological filament. However, there are no data in the manuscript that address whether the previously published single-stranded filament is, or is not, a physiological filament. Do the authors rule out that there could be two physiological architectures? The authors should include some analysis to compare the protofilaments. Given the very different twists, it is not obvious how single-stranded filaments (published before) would form from the salt-induced dissociation of this double-stranded filament – although this is difficult to judge without the coordinates.*

In order to address the first point we have tried to image crenactin filaments through electron tomography of *Pyrobaculum calidifontis* cells. No filaments were discernable. This might be because they are too close to the membrane where there are many other proteins (same situation as for MreB filaments) or filaments might be cell cycle or otherwise dependent on growth conditions. Overexpression in *E. coli* cells enabled us to visualise the filament in cells by tomography but that's not the real thing. Further investigations will be required. *Pyrobaculum calidifontis* is not genetically tractable right now, although it is very easy to grow.

The twist in the single protofilaments (cryoEM Braun et al., X-ray structures of single protofilament: Izoré et al. and Lindås et al) is reported to be around 45° from subunit to subunit. In the protofilament/single strand of double helical crenactin it is ~36°. While this is a big change, we concluded that because of the unclear relevance of the single protofilaments that this might not add much and potentially confuse. To make this more relevant, one might want to try to investigate if single filaments become double when conditions are changed from high-salt to low-salt, for example. We have not investigated this or if, as mentioned, double filaments become single when raising the salt concentration. It is just not clear if single filaments play any roles in cells and we suggest this is unlikely.

*8) Introduction, second paragraph: The statement that the MreB structure is the most divergent from F-actin structure is incorrect. There is plenty of evidence that ParMs have the most varied architectures.*

Many thanks for highlighting this issue. Two findings need to be separated: monomer structure and filament architecture (which itself can be split into longitudinal and lateral). Our statement meant to say that MreB filaments are the most unusual since they are non-helical, antiparallel and membrane-attached, unlike any other actin or actin-like filament. ParM's subunit structure is most diverged but the filaments are still double helical, polar and staggered. Text changed: "MreB, essential for cell-shape maintenance in most rod-shaped bacteria, forms the most divergent filaments when compared to F-actin, as it forms apolar, non-staggered and non-helical filaments that bind directly to membranes (Salje et al., 2011; van den Ent et al., 2014)."

*9) Figure 3—figure supplement 2 – These two figure panels are very important. They should be moved into the main paper. The direction of the sequestering helices in each case should be noted – they appear to vary.*

That's a good suggestion, changed the figure. There were actually other problems with the figure panels that have been rectified (ParM:ParR showed the wrong structure and tb4 used the wrong structure and reference). The termini pointing away from the viewer have been labelled on the proteins binding to the hydrophobic groove, highlighting the reversed orientation in the thymosin structure, which now comes from PDB 2PL8. Also, the reversed direction is now mentioned in the figure legend.